

# Establishment of an immortalized mouse dermal papilla cell strain with optimized culture strategy

Haiying Guo[1,*], Yizhan Xing[1,*], Yiming Zhang[2], Long He[1,3], Fang Deng[1], Xiaogen Ma[1] and Yuhong Li[1]

[1] Department of Cell Biology, Army Medical University, Chongqing, China
[2] Department of Plastic and Cosmetic surgery, Xinqiao Hospital, Army Medical University, Chongqing, China
[3] "111" Project Laboratory of Biomechanics and Tissue Repair & Key Laboratory of Biorheological Science and Technology of Ministry of Education, College of Bioengineering, Chongqing University, Chongqing, China
[*] These authors contributed equally to this work.

## ABSTRACT

Dermal papilla (DP) plays important roles in hair follicle regeneration. Long-term culture of mouse DP cells can provide enough cells for research and application of DP cells. We optimized the culture strategy for DP cells from three dimensions: stepwise dissection, collagen I coating, and optimized culture medium. Based on the optimized culture strategy, we immortalized primary DP cells with SV40 large T antigen, and established several immortalized DP cell strains. By comparing molecular expression and morphologic characteristics with primary DP cells, we found one cell strain named iDP6 was similar with primary DP cells. Further identifications illustrate that iDP6 expresses FGF7 and α-SMA, and has activity of alkaline phosphatase. During the process of characterization of immortalized DP cell strains, we also found that cells in DP were heterogeneous. We successfully optimized culture strategy for DP cells, and established an immortalized DP cell strain suitable for research and application of DP cells.

## INTRODUCTION

Hair follicles have the characteristic of periodical growth, which provides a nice model for the research of tissue regeneration. Dermal papilla (DP) cells have contact with hair follicle stem cells regularly and may play important roles in the regeneration of hair follicle (*Su et al., 2017*; *Woo et al., 2017*). The signals from DP may regulate the regeneration of hair follicles and melanocyte (*Guo et al., 2016*; *Li et al., 2013*). Dissociated human DP cells induce hair follicle neogenesis in grafted dermal-epidermal composites (*Thangapazham et al., 2014*). The limitation for DP research lies in the difficulty for culture of DP cells (*Morgan, 2014*). As so far, the human intact dermal papilla transcriptional signature can be partially restored by growth of papilla cells in 3D spheroid cultures (*Topouzi et al., 2017*). When the culture environment was changed into 2D environment, very rapid and profound molecular signature changes were discovered (*Higgins et al., 2013*; *Lin et al., 2016*). The isolation method of DP by surgical microdissection has been established in mouse vibrissae

Corresponding author
Yuhong Li,
liyuhongtmmu@hotmail.com

follicles and in human hair follicles (*Gledhill, Gardner & Jahoda, 2013*), but the isolated DP cells can not be long-term cultured. Since the isolation of primary DP cells is time-consuming and has limited population doubling. There are also several inter-individual and intra-individual variations. It is necessary to establish stable DP cell lines to investigate hair biology. Immortalized DP cell lines of human have been established, and had hair growth promoting effects (*Shin et al., 2011*; *Won et al., 2010*). In rodent animal models, immortalized rat DP cells have already been obtained (*Kang et al., 2015*). However, an effective immortalized mouse DP cell line is to be constructed. The goals of this project are optimize the isolation and culture condition of DP from mouse skin and establish an immortalized DP cell line for future research.

## MATERIALS AND METHODS

### Isolation and culture of DP cells

C57BL/6 mice were obtained from and housed in the laboratory animal center of the Army Medical University, Chongqing, China. All the animal-related procedures were conducted in strict accordance with the approved institutional animal care and maintenance protocols. All experimental protocols were approved by the Laboratory Animal Welfare and Ethics Committee of the Army Medical University. Permission number for producing animals: SCXK-PLA-20120011. Permission number for using animals: SYXK-PLA-20120031.

A 9-day old C57BL/6 mouse was sacrificed according to standard protocol. The vibrissa pads were cut off bilaterally with an iris scissor in a 100-mm plate. Vibrissa pads were rinsed with PBS, and then hair follicles were dissected together with their connective tissue sheath using 27G syringe needles under dissecting microscope. The dissected hair follicles were rinsed with PBS and incubated with 0.25% dispase for 20 min at room temperature.

Dissected hair follicles were transferred into a new 100-mm plate and thoroughly washed with PBS. A horizontal cut directly above dermal papilla was made. After that, dermal papilla was dissected out of dermal sheath using 27G syringe needles under dissecting microscope. Then the dissected DP tissues were transferred into a 10 $\mu g/cm^2$ collagen I coated 24-well plate. DP media were added after 30 min incubation in 37 °C. DP cells presented at about 3 days later. Cells reach confluence after 2 weeks and were passaged onto collagen I coated plates. DP medium should include α-MEM (Gibco, Waltham, MA, USA), 10% FBS (Gibco, USA), 1 × sodium pyruvate (Gibco, USA), 1 × non-essential amino acid (Gibco, USA), 1 × penicillin-streptomycin, 10 ng/ml bFGF (PeproTech, Rocky Hill, NJ, USA). During the optimization process, the classical DP medium was used as control. The control medium is consisted of DMEM (Gibco, USA), 10% FBS (Gibco, USA), 1 × penicillin-streptomycin. Another control medium was the classical DP medium with the addition of 10 ng/ml bFGF (PeproTech, USA).

### Establishment of immortalized DP cell line

Retrovirus with SV40 large T antigen which was flanked with FRT sites was prepared as formerly reported (*Yang et al., 2012*). Primary DP cells were plated in a 60 mm-dish at 50% confluency in the morning. After attachment, polybrene (final concentration 10 μg/mL) and retrovirus ($3.0 \times 10^7$ TU) were added together. The second day, the supernatant was

**Table 1** The information of primers in RT-PCR experiment.

| Primers | Sequence | Tm (°C) | Product size |
|---|---|---|---|
| Noggin-f | 5′ AGCACCCAGCGACAACCT 3′ | 61 | 343 bp |
| Noggin-r | 5′ CAGCCACATCTGTAACTTCCTC 3′ | | |
| Tbx18-f | 5′ GCTGCTAACCAGACCCAC 3′ | 58 | 537 bp |
| Tbx18-r | 5′ GTCCATGTCGCCAATACTC 3′ | | |
| Bmp6-f | 5′ TGCCTTAAACCACGAACAA 3′ | 58 | 345 bp |
| Bmp6-r | 5′ GCTGGGAATGGAACCTGAA 3′ | | |
| Fgf7-f | 5′ AGCGGAGGGGAAATGTTCG 3′ | 61 | 238 bp |
| Fgf7-r | 5′ TCCAGCCTTTCTTGGTTACTGAGA 3′ | | |
| Sostdc1-f | 5′ CCCCCATCCCAGTCATTTCTT 3′ | 58 | 308 bp |
| Sostdc1-r | 5′ CAGGGGGATAATTTCACACTGAGA 3′ | | |
| Sox2-f | 5′ AAAACCGTGATGCCGACTA 3′ | 58 | 431 bp |
| Sox2-r | 5′ ATCCGAATAAACTCCTTCCTTG 3′ | | |
| SMA-f | 5′ AGGGAGTAATGGTTGGAATGG 3′ | 58 | 351 bp |
| SMA-r | 5′ CATCTCCAGAGTCCAGCACAA 3′ | | |
| Gapdh-f | 5′ ACCACAGTCCATGCCATCAC 3′ | 52 | 450 bp |
| Gapdh-r | 5′ TCCACCACCCTGTTGCTGTA 3′ | | |

aspirated out of dish, and new DP medium was refilled. At the same time, hygromycin was added at a final concentration of 200 μg/mL. The culture medium was changed every two days until all the cells in control group died.

Antibiotic-selected DP cells were diluted with DP medium to 1–2 cells per 100 μL, and 100 μL diluted cells were added into every well of 96-well plates. Wells with only one cell were labeled and were monitored every 2 days. Cells in the labeled wells were passaged when the cell number of clones reached 20 or more.

## RT-PCR

Total RNA of immortalized DP cells were extracted with Eastep$^{TM}$ super total RNA extraction kit (Promega, Beijing, China) according to manufacturer's protocol. Complementary DNA was synthesized from RNA using Rever Tre Ace cDNA synthesis Kit (Toyobo, Osaka, Japan) according to the manufacturer's protocol. Several gene expressions were determined by PCR machine (Bio-Rad, Hercules, CA, USA) with the synthesized cDNA as template. The primers used were shown in Table 1. PCR mastermix (Novoprotein, Shanghai, China) were used when amplifying. The reannealing temperatures (Tm) and product size for the primers were also shown in Table 1.

## Immunocytochemistry staining

Cover slides were placed on a 24-well plate, and cells were plated on cover slides. Twenty-four hours later, cover slides were rinsed with PBS and fixed with acetone. Then, the cover slides were rinsed with PBS and incubated with 5% goat serum in PBS at room temperature for 1 h. After that, slides were incubated with a rabbit anti-FGF7 antibody (1:100; Boster, Wuhan, China) or a rabbit anti-α-SMA antibody (1:200; Bioss, Beijing, China) at 4 °C overnight and subsequently with appropriate secondary antibodies (1:500; ZSGB-bio,

Beijing, China). The slides were counterstained with DAPI (1:10,000; Beyotime, Shanghai, China) for 10 min. At last, the cover slides were moved to microscope slides, mounted with antifade mounting medium (Beyotime, Shanghai, China), and observed under fluorescent microscope. The immunostaining experiments were repeated three times.

### Alkaline phosphatase staining

Cover slides were placed on a 24-well plate, and cells were plated on cover slides. Twenty-four hours later, cover slides were rinsed with PBS and fixed with *in situ* fixation solution (Beyotime, Shanghai, China) for 10 min. Then the cover slides were rinsed with PBS five times. Fresh made NBT/BCIP staining buffer (Beyotime, Shanghai, China) or BM purple (Roche, Indianapolis, IN, USA) were added into the wells. The plate was covered with aluminium foil in the dark. Color change was monitored every 15 min to avoid non-specific staining. After the colour change appeared, the staining solution was aspirated out and the cells were washed twice with $1 \times$ PBS. At last, the cover slides were dehydrated, cleared, moved to microscope slides, mounted with permount (ZSGB-bio, Beijing, China), and observed under microscope. The AP staining experiments were performed twice.

### Detection of immortalization

Primary DP cells and iDP6 cells were cultured. The iDP6 cells were treated with AdGFP (adenovirus with the ability to express GFP protein), AdFlip (adenovirus with the ability to express flip recombinase, which can interact with FRT thus remove the expression of SV40) or PBS. Forty-eight hours later, cells were collected and total proteins were extracted with RIPA lysis buffer (Beyotime, Shanghai, China). Then, total proteins were loaded to 1% SDS-PAGE gel (Beyotime, China) and transmitted to PVDF membrane (Bio-Rad, Hercules, CA, USA). The PVDF membrane were incubated with anti-SV40 (1:1,000; Santa Cruz Biotechnology, Dallas, TX, USA) and anti-GAPDH (1:500; ZSGB-bio, Beijing, China) antibodies. HRP labelled secondary antibodies were used, and the results were observed under ChemiDoc$^{TM}$ Touch Imaging System (Bio-Rad, Hercules, CA, USA). The experiment on reversing immortalization was performed twice.

## RESULTS

### DP cells can be long-term cultured with the optimized strategy

We optimized the culture strategy for DP cells from three dimensions, plate coating, dissecting method, and culture media (Fig. 1). The optimized dissecting method worked well in obtaining primary DP cells. DP cells grew better on plate coated with collagen I than on uncoated plate. The morphology of DP cells did not have any significant difference between classical DP culture medium (DMEM with 10% FBS) and classical DP culture medium with the addition of bFGF (data not shown). Compared with classical DP culture medium, primary DP cells grew better in the optimized culture medium (Figs. 2A–2D). The morphology of passaged DP cells was much more resemble in primary DP cells in the optimized culture medium. The cultured DP cells still had the characteristics of agglutinative growth in the optimized culture medium, but not in the control medium (Figs. 2E–2H).

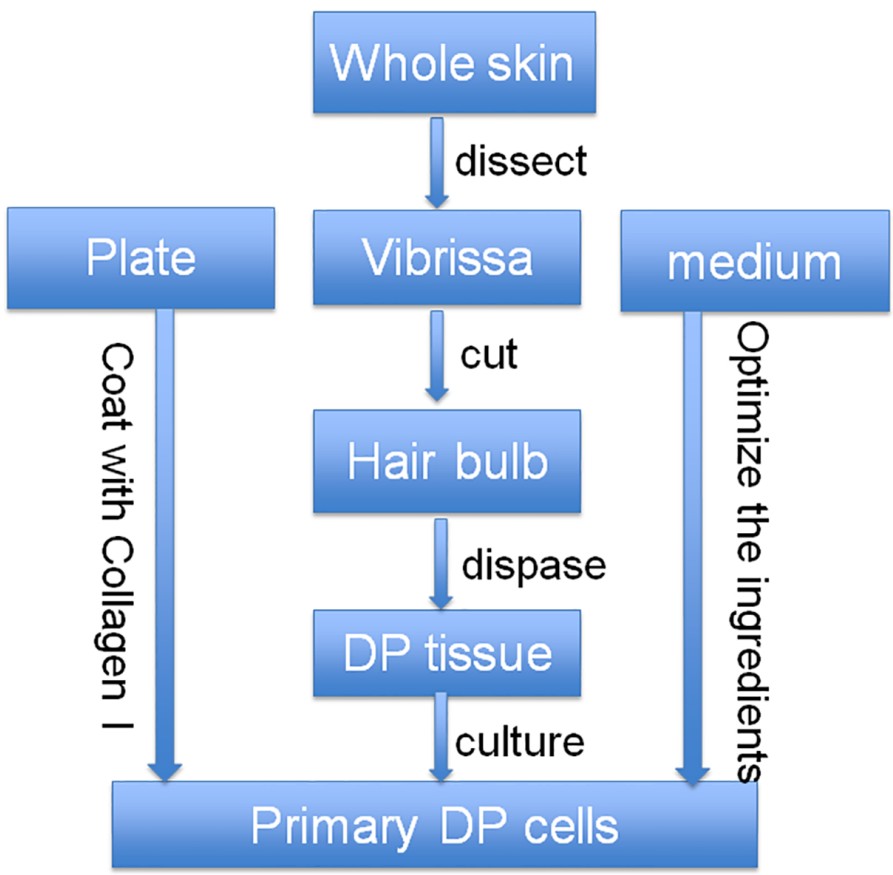

**Figure 1  Optimized strategy for the isolation and culture of DP cells.** At first, the whole skin of vibrissa area was cut, then the DP tissue was separated from the skin together with vibrissa pad, and then the DP tissue was collected after dispase digestion. After that, the collected DP tissue was cultured with our optimized culture medium in collagen I-coated plate.

## DP cells are heterogeneous

Primary DP cells were immortalized by SV40 system. DP cells before antibiotic-selection were named with 0#. After antibiotic-selection, DP cell strains were selected by infinite dilution method. Not every single cell grew to clone at last. Cell strains were named with the time sequence when they grew to clone beginning with just one single cell. Totally 19 cell strains survived at last, named with iDP1 to iDP19 (1#–19#). The morphologic characteristics of the selected cell strains were different from each other (Fig. 3). Some cells still look like fibroblast, whereas some cells changed into epithelial-like cells (Fig. 3G). iDP6 still had the characteristic of agglutinative growth, while others lost this characteristic. Specially, iDP10 grew clonally, which implied that the cell line was more primitive. For these cell strains, the expression patterns of the markers for DP cells were also determined by RT-PCR, including FGF7, BMP6, Sox2, Tbx18, Sostdc, α-SMA and noggin (Fig. 4). All these data indicate that the cell strains were totally different from each other. Since each cell line grew from one single DP cell, the DP cells from one DP tissue were heterogeneous.

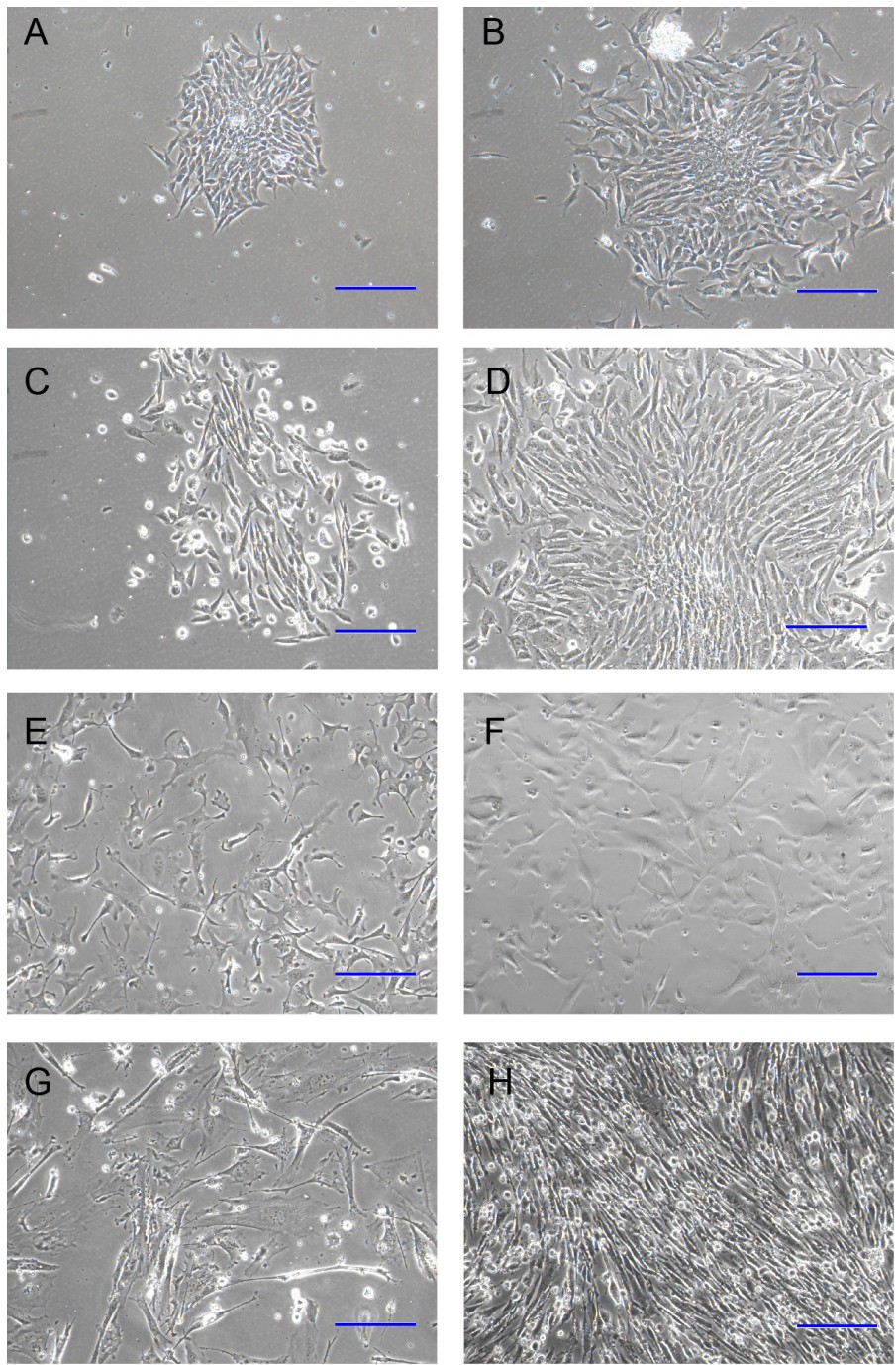

**Figure 2  Optimization of culture media for DP cells.** Cells in (A, C, E, G) are cultured in DMEM culture medium with 10% FBS, cells in (B, D, F, H) are cultured in optimized culture medium. (A)–(D) are primary DP cells. (A) and (B) are 2 days after culture; (C) and (D) are 4 days after culture. (E)–(H) are DP cells after one generation of passage. (E) and (F) are 2 days after passage; (G) and (H) are 4 days after passage (100×). Scale bar = 100 µm.

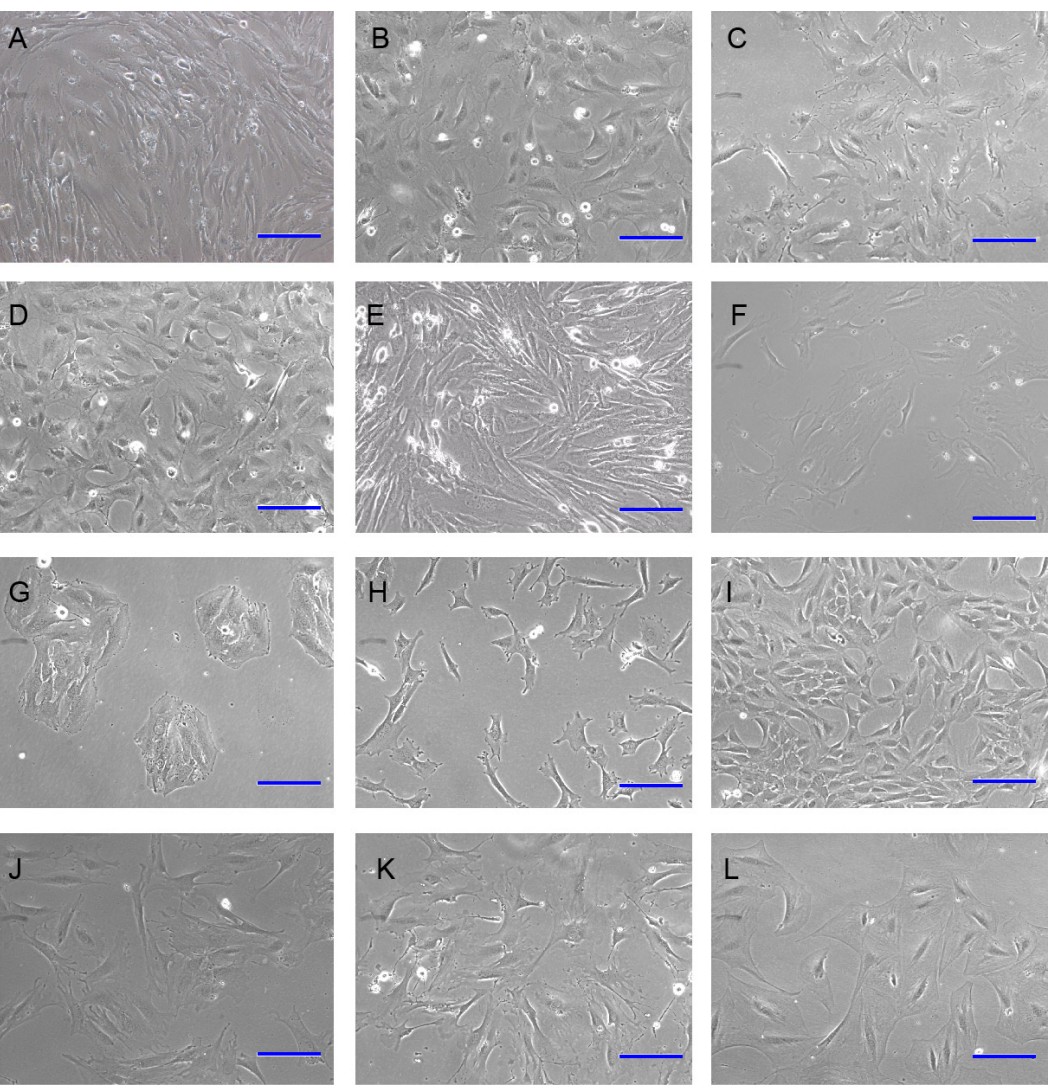

**Figure 3  Morphology of immortalized cell strains.** (A)–(L) represent cell strains named with 0#, 3#, 4#, 5#, 6#, 9#, 10#, 11#, 12#, 14#, 17#, 19#. Scale bar = 100 μm.

## iDP6 keeps the molecular characteristics of primary DP cells

Taken morphology and mRNA expression characteristics together, iDP6 is the one that most similar to the primary DP cells. To determine whether iDP6 can be used in DP research, the activity of alkaline phosphatase was determined by AP staining, and the expression of FGF7 and α-SMA were determined by immunocytochemistry as well. At protein level, just like *in situ*, some iDP6 cells still had high AP activity (Figs. 5A–5F). FGF7 was expressed in the cytoplasm of all iDP6 cells (Figs. 5G–5L). α-SMA was expressed in both the cytoplasm and the nucleus of all iDP6 cells (Figs. 5M–5R). Although the AP activity in iDP6 was lower than primary DP cells, the expression patterns of FGF7 and α-SMA was similar to primary DP cells.
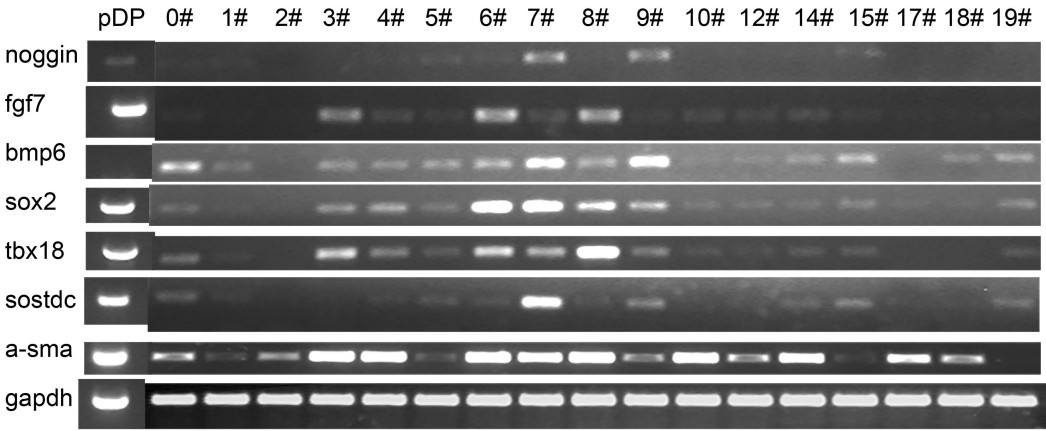

**Figure 4** **Expression pattern of the iDP6.** The expression of several known DP markers were determined by RT-PCR. GAPDH was used as internal control. Each lane represents a DP cell strain. pDP, primary DP cells.

### The immortal process of DP cells is reversible

At first, the expression of SV40 were determined to make sure that iDP6 were immortalized. Western blot showed that iDP6 cells expressed SV40, while primary DP cells did not express SV40 (Fig. 6A). Then, AdFlip was used to remove the expression of SV40 in iDP6 cells. AdGFP and PBS were used as control. Results showed that compared with control groups, the expression of SV40 decreased at 48 h after being treated with AdFlip (Fig. 6B). These results demonstrated that iDP6 was successfully immortalized and the immortal process was reversible.

## DISCUSSION

Primary cell culture needs to simulate the *in vivo* environment of the cells. In anagen, DP cells reside in the center of hair bulb. They are circumstanced with a single layer of dermal cells. Usually, DP cells periodically interact with epithelial cells outside of the single layer of dermal cells. In telogen, DP tissue is a little far from hair follicle stem cells (HFSCs). In anagen onset, it begins to move close to HFSCs, and keeps interaction with HFSCs during early anagen. In the late anagen, it begins to move away from HFSCs. In catagen, it keeps away from HFSCs. Thus, the environment for DP cells *in vivo* varies with hair cycle (*Bassino et al., 2015*). In addition, exogenous connective tissue may also impact the function of DP cells (*Zhang et al., 2014b*). To exclude contamination from other mesenchymal cells, epithelial cells, and adipose tissue, the use of microdissection techniques is preferred (*Zhang et al., 2014a*). To culture DP cells *in vitro*, all the conditions should be taken into consideration. As the main component of connective tissue, collagen is mostly secreted by fibroblast. Collagen is widely used in tissue engineering and cell culture. We coated the plate with collagen, and found it was good for the growth of DP cells. DP is relatively independent in the anagen hair follicle. However, it is too small to be isolated quickly. So we used a stepwise method to isolate DP. DP cells grew from the isolated DP.

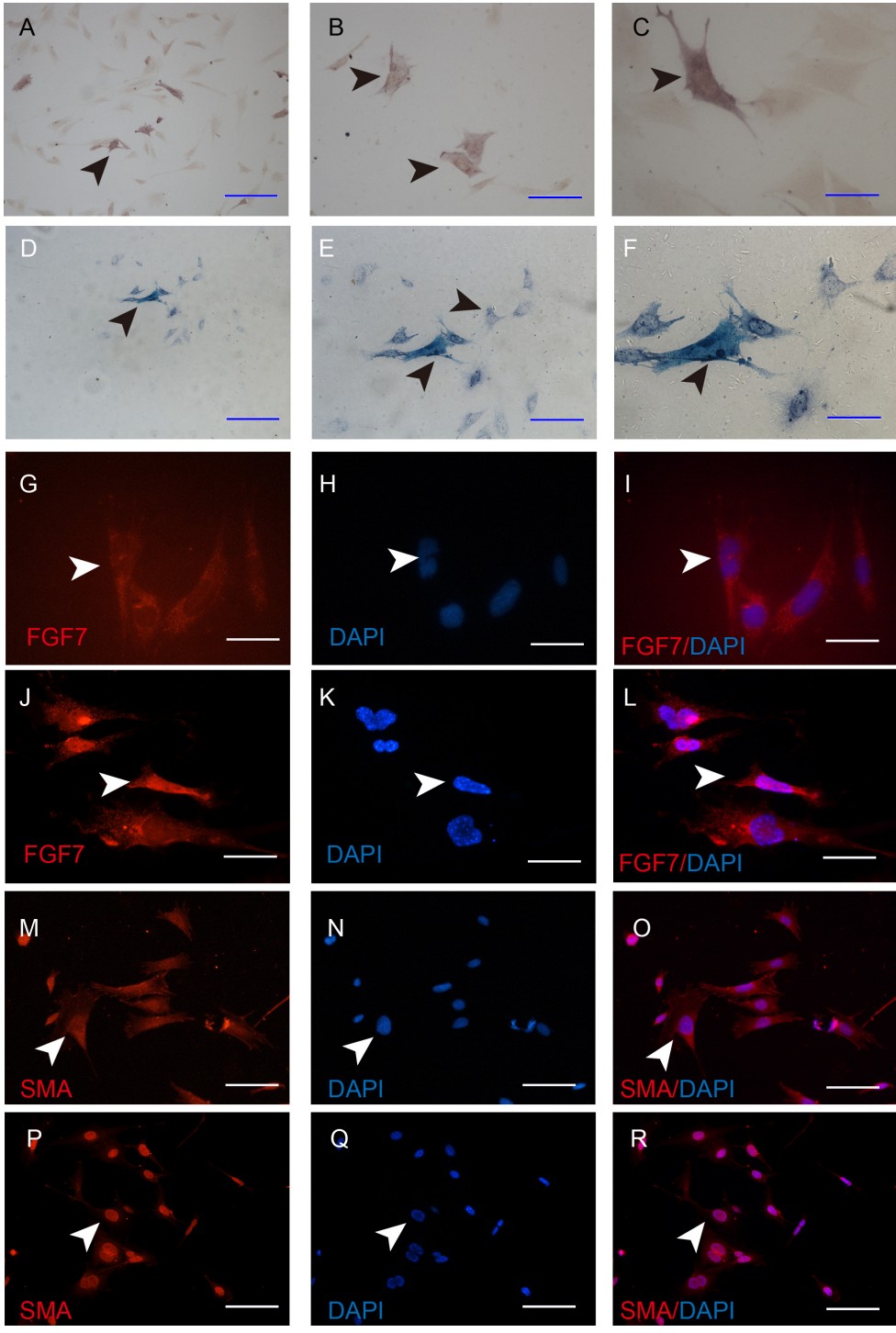

**Figure 5** **Characterization of the iDP6.** (A)–(C), (G)–(I), (M)–(N) Characterization of iDP6. (D)–(F), (J)–(L), (P)–(R) Characterization of primary DP cells. (A)–(F) Alkaline phosphatase staining. (G)–(R) Immunocytochemistry, red color denotes positive expression, blue color denotes the counterstaining of DAPI. (G)–(R) The right panels are the merge of the left two panels. Arrowheads denote the positive expression. Scales bars for (A) and (D) are 100 μm. Scale bars for (B), (E), and (M)–(R) are 50 μm. Scale bars for (C), (F) and (G)–(L) are 25 μm.

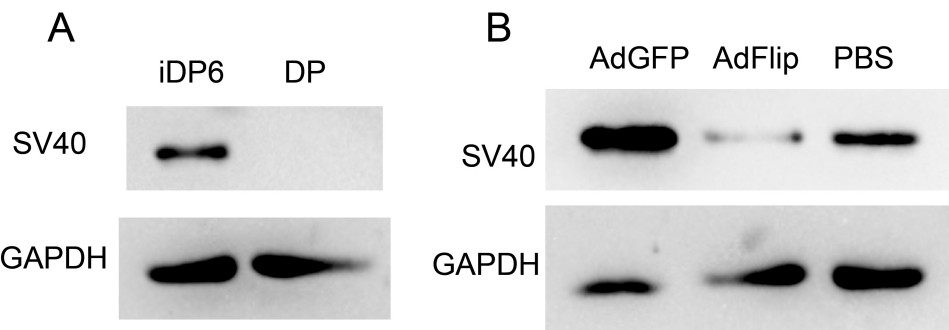

**Figure 6** **Reversible immortalization of DP cells.** The expressions of SV40 large T antigen were determined by western blot. (A) The expression of SV40 in iDP6 and primary DP. (B) The expression of SV40 in adenovirus treated iDP6. DP, primary dermal papilla cells; iDP6, immortalized DP cells #6; AdGFP, Ad-Flip; PBS, iDP6 cells treated with AdGFP, AdFlip or PBS. Proteins were collected at 48 h after treatment.

The most important condition for cell culture is the culture medium. Since the classical DMEM with 10% FBS can not long-term culture mouse DP cells, we seek to find culture medium for special fibroblast. We found that a mesenchymal stem cell culture medium α-MEM worked well. Additionally, bFGF is a critical component of human embryonic stem cell culture medium. In conjunction with BMP4, bFGF promotes differentiation of stem cells to mesodermal lineages (*Yuan et al., 2013*). DP originates from mesodermal, so we added bFGF to the medium. However, since the classical medium and the classical medium with the addition of bFGF did not have significant difference on the culture of primary DP, the main effective additions in the optimized medium maybe sodium pyruvate and non-essential amino acids. Based on these data, the optimized strategy works well in isolating and long-term culture of DP cells. We are the first to use this strategy to culture DP cells.

There are several molecules reported to be expressed in DP cells, including Sox2, Tbx18, Sostdc, α-SMA and noggin (*Weber & Chuong, 2013*). To characterize immortalized DP cells, all the markers were tested. Recently, we found that two secretive proteins, FGF7 and BMP6 were also expressed in DP cells *in vivo*. FGF signaling was reported to regulate the size of dermal papilla (*Yue et al., 2012*), and BMP7 was reported to attenuate fibroblast-like differentiation of DP cells (*Bin et al., 2013*). Thus the expressions of FGF7 and BMP6 are also tested. Both the expressions of markers and morphology indicate that the immortalized DP cell strains are heterogeneous and iDP6 is a good cell strain to represent primary DP cells. It is reported that human DP cells have stem cell-like phenotypes (*Kiratipaiboon, Tengamnuay & Chanvorachote, 2016*), neural crest stem cell-like cells were also isolated from rat vibrissa DP (*Li et al., 2014*), dermal stem cells also lies in mouse dermal sheath (*Rahmani et al., 2014*). So it is reasonable that DP cells are heterogeneous. But exactly how many kinds of cells are in DP remain to be discovered (*Yang et al., 2017*). Single cell assay technologies may help.

## CONCLUSIONS

From the results of present study, it can be concluded that we optimized the dissection and culture of mouse DP cells from three dimensions: stepwised dissection, collagen I coated plate and α-MEM based culture medium. Based on the optimized strategies, we successfully immortalized the cultured primary DP cells with addition of SV40 large T antigen. We successfully selected several cell strains, characterized them, and found iDP6 cell strain similar to primary DP cells. In addition, the SV40 large T antigen in iDP6 can be removed by the addition of AdFlip. In a word, we establised an immortalized DP cell strain that can be used in future research.

## ACKNOWLEDGEMENTS

We thank Tong-chuan He in Chicago University for providing the SV40 large T antigen associated plasmids. We thank Ke Yang in Chongqing Medical University for helping in the immortalization experiments. We thank Claire Higgins and other reviewers for constructive suggestions on the revision of the manuscript.

### Funding

This work was supported by the National Natural Science Foundation of China (No. 81472895) and the Natural Science Foundation of Chongqing (No. cstc2015jcyjA1219). There was no additional external funding received for this study. The funders had no role in study design, data collection and analysis, decision to publish, or preparation of the manuscript.

### Grant Disclosures

The following grant information was disclosed by the authors:
National Natural Science Foundation of China: 81472895.
Natural Science Foundation of Chongqing: cstc2015jcyjA1219.

### Competing Interests

The authors declare there are no competing interests.

### Author Contributions

- Haiying Guo, Yizhan Xing and Yiming Zhang performed the experiments, reviewed drafts of the paper.
- Long He performed the experiments, contributed reagents/materials/analysis tools, reviewed drafts of the paper.
- Fang Deng contributed reagents/materials/analysis tools, reviewed drafts of the paper.
- Xiaogen Ma analyzed the data, reviewed drafts of the paper.
- Yuhong Li conceived and designed the experiments, wrote the paper, prepared figures and/or tables, reviewed drafts of the paper.

## Animal Ethics

The following information was supplied relating to ethical approvals (i.e., approving body and any reference numbers):

All experimental protocols were approved by the Laboratory Animal Welfare and Ethics Committee of the Army Medical University. Permission number for producing animals: SCXK-PLA-20120011, Permission number for using animals: SYXK-PLA-20120031.

## Data Availability

Li, Yuhong (2018): raw data for PeerJ-R1.zip. figshare. 10.6084/m9.figshare.5752476.

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

# PeerJ

**Lin B, Miao Y, Wang J, Fan Z, Du L, Su Y, Liu B, Hu Z, Xing M. 2016.** Surface tension guided hanging-drop: producing controllable 3D spheroid of high-passaged human dermal papilla cells and forming inductive microtissues for hair-follicle regeneration. *ACS Applied Materials & Interfaces* **8**:5906–5916 DOI 10.1021/acsami.6b00202.

**Morgan BA. 2014.** The dermal papilla: an instructive niche for epithelial stem and progenitor cells in development and regeneration of the hair follicle. *Cold Spring Harbor Perspectives in Medicine* **4**:Article a015180 DOI 10.1101/cshperspect.a015180.

**Rahmani W, Abbasi S, Hagner A, Raharjo E, Kumar R, Hotta A, Magness S, Metzger D, Biernaskie J. 2014.** Hair follicle dermal stem cells regenerate the dermal sheath, re-populate the dermal papilla, and modulate hair type. *Developmental Cell* **31**:543–558 DOI 10.1016/j.devcel.2014.10.022.

**Shin SH, Park SY, Kim MK, Kim JC, Sung YK. 2011.** Establishment and characterization of an immortalized human dermal papilla cell line. *BMB Reports* **44**:512–516 DOI 10.5483/BMBRep.2011.44.8.512.

**Su YS, Fan ZX, Xiao SE, Lin BJ, Miao Y, Hu ZQ, Liu H. 2017.** Icariin promotes mouse hair follicle growth by increasing insulin-like growth factor 1 expression in dermal papillary cells. *Clinical and Experimental Dermatology* **42**:287–294 DOI 10.1111/ced.13043.

**Thangapazham RL, Klover P, Wang JA, Zheng Y, Devine A, Li S, Sperling L, Cotsarelis G, Darling TN. 2014.** Dissociated human dermal papilla cells induce hair follicle neogenesis in grafted dermal-epidermal composites. *Journal of Investigative Dermatology* **134**:538–540 DOI 10.1038/jid.2013.337.

**Topouzi H, Logan NJ, Williams G, Higgins CA. 2017.** Methods for the isolation and 3D culture of dermal papilla cells from human hair follicles. *Experimental Dermatology* **26**:491–496 DOI 10.1111/exd.13368.

**Weber EL, Chuong CM. 2013.** Environmental reprogramming and molecular profiling in reconstitution of human hair follicles. *Proceedings of the National Academy of Sciences of the United States of America* **110**:19658–19659 DOI 10.1073/pnas.1319413110.

**Won CH, Choi SJ, Kwon OS, Park WS, Kang YJ, Yoo HG, Chung JH, Cho KH, Eun HC, Kim KH. 2010.** The establishment and characterization of immortalized human dermal papilla cells and their hair growth promoting effects. *Journal of Dermatological Science* **60**:196–198 DOI 10.1016/j.jdermsci.2010.08.015.

**Woo H, Lee S, Kim S, Park D, Jung E. 2017.** Effect of sinapic acid on hair growth promoting in human hair follicle dermal papilla cells via Akt activation. *Archives of Dermatological Research* **309**:381–388 DOI 10.1007/s00403-017-1732-5.

**Yang H, Adam RC, Ge Y, Hua ZL, Fuchs E. 2017.** Epithelial-Mesenchymal micro-niches govern stem cell lineage choices. *Cell* **169**:483–496.e13 DOI 10.1016/j.cell.2017.03.038.

**Yang K, Chen J, Jiang W, Huang E, Cui J, Kim SH, Hu N, Liu H, Zhang W, Li R, Chen X, Kong Y, Zhang J, Wang J, Wang L, Shen J, Luu HH, Haydon RC, Lian X, Yang T, He TC. 2012.** Conditional immortalization establishes a repertoire of mouse melanocyte progenitors with distinct melanogenic differentiation potential. *Journal of Investigative Dermatology* **132**:2479–2483 DOI 10.1038/jid.2012.145.

**Yuan S, Pan Q, Fu CJ, Bi Z. 2013.** Effect of growth factors (BMP-4/7 & bFGF) on proliferation & osteogenic differentiation of bone marrow stromal cells. *Indian Journal of Medical Research* **138**:104–110.

**Yue Z, Jiang TX, Wu P, Widelitz RB, Chuong CM. 2012.** Sprouty/FGF signaling regulates the proximal-distal feather morphology and the size of dermal papillae. *Developmental Biology* **372**:45–54 DOI 10.1016/j.ydbio.2012.09.004.

**Zhang P, Kling RE, Ravuri SK, Kokai LE, Rubin JP, Chai JK, Marra KG. 2014a.** A review of adipocyte lineage cells and dermal papilla cells in hair follicle regeneration. *Journal of Tissue Engineering* **5**:Article 2041731414556850 DOI 10.1177/2041731414556850.

**Zhang P, Ravuri SK, Wang J, Marra KG, Kling RE, Chai J. 2014b.** Exogenous connective tissue growth factor preserves the hair-inductive ability of human dermal papilla cells. *International Journal of Cosmetic Science* **36**:442–450 DOI 10.1111/ics.12146.