# Peer review of "Establishment of an immortalized mouse dermal papilla cell strain with optimized culture strategy"

_PeerJ, doi:10.7717/peerj.4306_

## Round 0.1 · original submission · Major Revisions

Please pay particular attention to how you are referring to the 2D and 3D environment of the cells in culture, and how this relates to their in vivo situation. As is, your text risks significantly confusing the reader. Also, when referring to improvements in DP isolation.and in development of a superior medium, these elements need to be much more explicitly stated in a way that makes them clearly distinct from other methods in the literature (which need to be cited).

Reviewer 1 ·

Basic reporting

Relevant prior literature on human immortalized human dermal papilla cell line should be referenced. For example, Shin et al., Establishment and characterization of an immortalized human dermal papilla cell. BMB Rep. 2011 Aug;44(8):512-6. Also, Won et al., The establishment and characterization of immortalized human dermal papilla cells and their hair growth promoting effects. J Dermatol Sci. 2010 Dec;60(3):196-8.

Experimental design

No comment

Validity of the findings

No comment

·

Basic reporting

The authors have clearly referred to relevant literature. However, sometimes the prose is confusing. For example in the introduction the phrase 'when the culture system was changed to 2D' is misleading. This suggests that intact DP which are being placed in 2D are in a culture system as well, which is not the case. The idea that intact DP are in a 3D culture system is also suggested within the paper, which again is incorrect. Cells within intact DP are in a 3D environment, but this is not a culture environment. The formatting and structure of sentences should be checked to ensure they are not misleading.
There are also numerous typos in the manuscript-eg Western Bloting rather than Blotting.

Experimental design

The aim of the paper is to 'optimise isolation and culture' methods of DP and 'establish an immortalised line'. The reason for establishing an immortalised line is not clear-suggestions as to why an immortalised line would be beneficial over a primary line would help here.
One of the reasons for performing this work is that DP cells are difficult to isolate, however the method of isolation in the paper appears the same as that described within the literature. Please explain how the isolation method is changed to make DP isolation easier?
n numbers should be included-i.e. how many times were the immunostaining protocols performed on cells. How many times was the immortalisation reversing protocol attempted. Did it work on multiple iDP lines?
The control media used is dMEM with 10% FBS, however this is media commonly used for human DP cells. Mouse DP cells (which are being assessed here) are usually grown in dMEM with 10% FBS, with additional growth factors such as FGF2. Why did the authors not have murine DP media as the control medium?
Temperatures used for PCR should be indicated. Product size should also be indicated in Table 1.

Validity of the findings

One of the main results of the paper is that the authors optimised a media to culture DP cells. Optimisation suggests that several media were attempted, but these are not shown. How many other medias were attempted, and what enabled the conclusion that the selected one was the best. In addition, please explain how the optimised media is different from that which is usually used.
Another finding in the paper is that iDP6 is the optimal induced line as it has similarity to primary DP cells. However, the profile of primary cells is not shown. To make this argument more convincing, the PCRs and staining (Figures 4/5) should include DP cells which are not immortalised.
What do the authors mean by 'agglutination'? Please add a sentence to explain.

---

## Round 0.2 · Minor Revisions

The reviewers and I feel that you have made a good effort to improve the article, but that there remains some minor issues to be addressed. Please focus on the reviewer's comments herein.

·

Basic reporting

N/A

Experimental design

Methods are much better, with n numbers and more detail on the experiments performed.

Validity of the findings

N/A

Additional comments

It seems from the expanded methods, and the response to reviewers that the main addition to the media, was sodium pyruvate and non-essential amino acids, and FGF2 had little effect by itself. This should be reflected in the discussion, especially page 6, lines 12-16.

page 6, line 28-add the following reference after 'But exactly how many kinds of cells are in DP remain to be discovered'
Epithelial-Mesenchymal Micro-niches Govern Stem Cell Lineage Choices.
Yang H, Adam RC, Ge Y, Hua ZL, Fuchs E.
Cell. 2017 Apr 20;169(3):483-496.e13. doi: 10.1016/j.cell.2017.03.038.

I wouldn't call primary DP cells 'wild-type'. This makes it seem like the others were from a transgenic/mutant mouse. Just call them primary. Update text and the figure legends for Figure 4/5 accordingly.

---

## Round 0.3 · accepted · Accept

Thank you for paying close attention to the reviewers' comments and responding accordingly.